# Acid Rain Increases Impact of Rice Blast on Crop Health via Inhibition of Resistance Enzymes

**DOI:** 10.3390/plants9070881

**Published:** 2020-07-13

**Authors:** Hong-Ru Li, Hui-Min Xiang, Jia-Wen Zhong, Xiao-Qiao Ren, Hui Wei, Jia-En Zhang, Qiu-Yuan Xu, Ben-Liang Zhao

**Affiliations:** 1Guangdong Provincial Key Laboratory of Eco-Circular Agriculture, South China Agricultural University, Guangzhou 510642, China; lihongru19@stu.scau.edu.cn (H.-R.L.); hmxiang@scau.edu.cn (H.-M.X.); zhongjiawen@stu.scau.edu.cn (J.-W.Z.); renxq@stu.scau.edu.cn (X.-Q.R.); weihui@scau.edu.cn (H.W.); xuqy@scau.edu.cn (Q.-Y.X.); blzhao@scau.edu.cn (B.-L.Z.); 2Department of Ecology, College of Natural Resources and Environment, South China Agricultural University, Guangzhou 510642, China; 3Key Laboratory of Agro-Environment in the Tropics, Ministry of Agriculture and Rural Affairs, South China Agricultural University, Guangzhou 510642, China; 4Guangdong Engineering Research Center for Modern Eco-Agriculture and Circular Agriculture, Guangzhou 510642, China; 5Key Laboratory of Agroecology and Rural Environment of Guangdong Regular Higher Education Institutions, South China Agricultural University, Guangzhou 510642, China

**Keywords:** acid rain, *Oryza sativa* L., *Pyricularia oryzae*, antioxidant enzyme activity, metabolism, disease index

## Abstract

Worldwide, rice blast (*Pyricularia oryzae*) causes more rice crop loss than other diseases. Acid rain has reduced crop yields globally for nearly a century. However, the effects of acid rain on rice-*Pyricularia oryzae* systems are still far from fully understood. In this study, we conducted a lab cultivation experiment of *P. oryzae* under a series of acidity conditions as well as a glasshouse cultivation experiment of rice that was inoculated with *P. oryzae* either before (*P.* + SAR) or after (SAR + *P.*) simulated acid rain (SAR) at pH 5.0, 4.0, 3.0 and 2.0. Our results showed that the growth and pathogenicity of *P. oryzae* was significantly inhibited with decreasing pH treatments in vitro culture. The SAR + *P.* treatment with a pH of 4.0 was associated with the highest inhibition of *P. oryzae* expansion. However, regardless of the inoculation time, higher-acidity rain treatments showed a decreased inhibition of *P. oryzae* via disease-resistance related enzymes and metabolites in rice leaves, thus increasing disease index. The combined effects of high acidity and fungal inoculation were more serious than that of either alone. This study provides novel insights into the effects of acid rain on the plant–pathogen interaction and may also serve as a guide for evaluating disease control and crop health in the context of acid rain.

## 1. Introduction

Global food security depends on rice yields, which are significantly negatively impacted each year by blast disease *Pyricularia oryzae* [1,2,3]. *P. oryzae* invades rice leaf cells and colonizes adjacent cells through plasmodesmata in two to six days after inoculation [4]. It reduces the green leaf area and the photosynthesis, suppresses the plant protective system and overall plant metabolism, and eventually leads to leaf death [5].

Acid rain is a serious worldwide environmental issue [6]. China, a large agricultural country, now has the third largest acid rain zone in the world, following North America and Europe [7]. There is more than 40% of terrestrial area experiencing acid rain in China [8]. The safety of crop ecosystems faces significant challenges because of the negative impacts of acid rain on plant growth and health through injuring leaf structure, destructing cell membrane integrity, degrading chlorophyll, and causing disorder to the antioxidant system and physiological metabolism [9,10].

In the field, plants are rarely exposed to a single hazardous stress. Most agroforestry areas are threatened by both acid rain and disease [11]. Additive, antagonistic or synergistic interactions between acid rain pollution and pathogens on plants have been reported [12]. Many studies have focused on either the effects of acid rain or *P. oryzae* on plants [13,14,15], but the interactive effects of acid rain and *P. oryzae* on rice are still not clear [16]. Acid rain pollution could have effects on plant diseases by altering physiological metabolism and/or disease resistance systems of the host [17,18]. The antioxidant system appears especially sensitive to acid rain [10,19,20]. Previous studies have shown that *P. oryzae* combined with acid rain can induce rice plants to activate the defense system, improve the activity of plant peroxidases (POD) and phenylalanine ammonia-lyase (PAL), and stimulate the production of polyphenolic compounds, thus improving the plant resistance to stress [21,22]. Another research reported that mild acid rain at pH > 4.5 decreases plant resistance to pathogens and increases the disease index of falling needle disease and red leaf disease, but acid rain at pH < 4.5 reduces the disease index of *Pinus massoniana* [23]. Elsewhere, research has shown that the disease of *Pseudomonas. phaseolicola* was aggravated when it was inoculated after simulated acid rain treatment but was alleviated when it is inoculated before simulated acid rain [24]. Overall, the impacts of acid rain on disease indices of plants depend on the acidity and the temporal sequence of pollutant events and initial infection by the pathogen.

Acid rain alters the occurrence of plant diseases by influencing the growth and development of fungi and their virulence or aggressiveness on the surface of rice leaves. When the leaves infected by pathogenic fungi are exposed to acid rain, the production, survival, propagation, invasiveness, and virulence of the fungal conidia are affected by the acidity of acid rain [25]. For instance, weak acid rain has no effect on the growth of *Puccinia striiformis* f. sp. *tritici*, but, at pH ≤ 4.0, it significantly inhibited the growth of *Puccinia striiformis* f. sp. *tritici* [23]. Another study reported that acid rain significantly increased the incidence of *Botrytis Cinerea* [26], however, it did not affect the incidence of potato late blight [27]. These countervailing results suggest the response of the fungal communities on the surface of leaves to acid rain depends on the pH of the acid rain and the type of microorganism. Given the prevalence of acid rain and its adverse consequences, it is important to continue investigating whether and how acid rain affects individual plant disease systems, such as the rice-*P. oryzae* system.

The purpose of this study is to improve our understanding of a rice disease system under acid rain stress and verify whether acid rain has an inhibitory, neutral, or positive effect on the development of rice blast. We conducted an in vitro laboratory culture experiments to detect the growth of *P. oryzae* under acid stress and a series of simulated acid rain treatments (SAR) on a greenhouse rice-*P. oryzae* system using different inoculation times (either before or after each acid treatment). For better predictions regarding the tolerance mechanisms behind the observed outcomes, we measured *P. oryzae* growth traits and plant traits, the latter included morphological, antioxidant enzymes activities, metabolism and disease index. We hypothesized that acid rain can alter the plant disease index by changing the infectivity of *P. oryzae* and inducing the rice plant’s physiological response and defense process to external stress. 

## 2. Results

### 2.1. Effects of Acid Treatment on P. oryzae Colony Traits

Colony diameter of *P. oryzae* was significantly inhibited under the acid treatments relative to that in the control (*P* < 0.05), except at pH 5.0 treatment (*P* > 0.05, Figure 1A). At the end of cultivation, the final colony diameter of *P. oryzae* with different treatments showed a changing order as pH 7.0 (CK) > pH 4.0 > pH 3.0 > pH 2.0 (*P* < 0.05, Figure 1A).

The trend of *P. oryzae* growth rate was significantly different among the treatments. The growth rate of *P. oryzae* increased first and then decreased after the ninth day for colony under CK and pH 5.0. While the growth rates were relatively stable after the ninth and sixth day in the pH 4.0 and pH 3.0 treatments, respectively. In the pH 4.0 and pH 3.0 treatments, the growth rate was significantly lower than that in the CK during the incubation stage (*P* < 0.05, Figure 1B). *P. oryzae* did not grow under pH 2.0 treatment (Figure 1B).

Melanin content of *P. oryzae* was significantly higher at pH 5.0 treatment than that of the control, but was lower at pH 4.0, pH 3.0 and pH 2.0 treatments than that of control (*P* < 0.05, Figure 2), with decreases of 11.30%, 50.67% and 94.43%, respectively.

### 2.2. Effects of SAR and P. oryzae on the Total Number of Leaves 

The interaction between SAR treatments and inoculation of *P. oryzae* significantly affected the total number of leaves (*P* < 0.001, Figure 3). When treated with simulated acid rain only, compared with pH 7.0, the total number of leaves decreased at pH 3.0 and pH 2.0 treatments, lowest at pH 2.0 (*P* < 0.05). In the control group (pH 7.0), the total number of leaves reduced in both the spraying acid rain at pre-infection stage of *P. oryzae* (SAR + *P.*) and spraying acid rain at symptomatic stage of *P. oryzae* (*P.* + SAR) treatments, respectively, compared with the control. The total number of leaves under the *P.* + SAR treatment significantly reduced with decreasing pH value (*P* < 0.05). Under the treatment of SAR + *P.*, the total number of leaves increased by 18.85% at pH 4.0, but declined by 24.56% and 41.54% at pH 3.0 and pH 2.0, respectively, compared with those at pH 7.0. The total number of leaves significantly decreased by 41.29% in *P*. + SAR treatment, compared with that in SAR + *P*. treatment at pH 4.0 (*P* < 0.05). 

### 2.3. Disease Resistance-Related Enzyme Activities in Rice Leaves 

Activities of PAL, POD and Chitinase (CHT) in rice leaves under the SAR treatment and the two different inoculation treatments of *P. oryzae* are shown in Figure 4. When treated with acid rain only, the activities of PAL and CHT significantly increased, compared with pH 7.0. POD activity decreased by 23.67% at pH 2.0 but increased by 27.42%, 83.20% and 57.43% at pH 5.0, pH 4.0 and pH 3.0, respectively, compared with pH 7.0. In the control group (pH 7.0), the activities of PAL, POD and CHT in rice leaves increased in SAR + *P.* and *P.* + SAR, relative to the non-inoculated treatment.

PAL activity in SAR + *P.* treatments significantly increased by 24.24% and 22.22% at pH 5.0 and pH 4.0 respectively, compared with the non-inoculated treatments (*P* < 0.05). However, the PAL activity in the SAR + *P.* treatment significantly decreased by 10.53% at pH 2.0, relative to the non-inoculated treatment. Under the *P.* + SAR treatments, the PAL activity increased only at pH 5.0 treatment and decreased at pH 3.0 and pH 2.0 treatments, compared with the non-inoculated treatments (*P* < 0.05).

Decreases of POD activity were found in combined treatments of *P. oryzae* and SAR, compared with the non-inoculated treatments at pH 5.0, pH 4.0, pH 3.0 and pH 2.0. Under SAR + *P.* treatment, the POD activity significantly increased by 21.37% at pH 4.0 and decreased by 61.73% at pH 2.0 relative to non-inoculated treatment at pH 7.0 (*P* < 0.05). The POD activity was significantly higher in SAR + *P.* treatment than in *P.* + SAR treatment at pH 4.0 and pH 3.0 (*P* < 0.05).

In SAR + *P.* treatments, CHT activity significantly increased by 18.18% and 22.22% at pH 5.0 and pH 4.0, but significantly decreased by 11.36% and 10% at pH 3.0 and pH 2.0, respectively, compared with the corresponding non-inoculated treatments. Under the *P.* + SAR treatments, the CHT activity in rice leaves significantly decreased at pH 4.0, pH 3.0 and pH 2.0 by 16.67%, 22.45% and 35% respectively, but significantly increased by 18.18% at pH 5.0, compared with the corresponding non-inoculated treatments.

SAR and inoculation of *P. oryzae* showed significant interaction on the activities of all enzymes (*P* < 0.001, Figure 4). The results indicated that combined stress of SAR and *P. oryzae* treatment had an antagonistic effect on protective enzyme system.

### 2.4. Contents of Metabolism Compounds in Rice Leaves 

The metabolite content in rice leaves responded idiosyncratically to the SAR and inoculation of *P. oryzae* treatments (Table 1). A two-way ANOVA showed that the SAR and inoculation time exerted a significant interactive effect on the contents of metabolites in rice leaves (*P* < 0.001). When treated only with acid rain, the content of flavonoids, total phenol and soluble sugar in rice leaves increased significantly with decreasing of pH value, but lignin content showed no change at pH 5.0, pH 4.0 and decreased at pH 3.0, pH 2.0, compared with the pH 7.0 control (*P* < 0.05). At pH 7.0, except lignin, the contents of flavonoids, total phenol and soluble sugar were higher in the SAR + *P.* and *P.* + SAR treatments than in the non-inoculated treatment.

Under the SAR + *P.* treatment, the contents of flavonoids and total phenol at pH 4.0 elevated by 14.06% and 24.60%, respectively, but decreased by 29.71% and 17.57% at pH 2.0, respectively, relative to the corresponding treatment at pH 7.0. Under the *P.* + SAR treatment, the content of flavonoids and total phenol at pH 2.0 reduced by 31.63 % and 26.05%, respectively, compared with the corresponding treatment at pH 7.0. The flavonoids content was higher in SAR + *P.* than in *P.* + SAR for the pH 4.0 and pH 3.0 treatments (*P* < 0.05). The content of total phenol was significantly higher in SAR + *P.* treatments than in *P.* + SAR treatments at pH 5.0, pH 4.0 and pH 2.0 (*P* < 0.05). 

Lignin content in the SAR + *P.* treatments at pH 5.0 and pH 4.0 had no difference compared with that at pH 7.0, but decreased by 38.87% at pH 3.0 and 52.68% at pH 2.0 (*P* < 0.05). The lignin content decreased in the *P.* + SAR treatments at pH 4.0, pH 3.0 and pH 2.0 by 25.20%, 51.51% and 64.81%, respectively, compared with the corresponding treatment at pH 7.0. The content of lignin was significantly lower in *P.* + SAR treatments than in SAR + *P.* treatments at pH 4.0, pH 3.0 and pH 2.0 (*P* < 0.05) and the decreases were 28.21%, 20.67% and 25.62%, respectively. 

Compared with the corresponding treatment at pH 7.0, the soluble sugar content increased by 49.72% at pH 4.0 and decreased by 18.60% at pH 2.0 in SAR + *P.* treatments. Soluble sugar contents in *P.* + SAR treatments at pH 3.0 and pH 2.0 decreased significantly, compared with the corresponding treatment at pH 7.0. In the pH 2.0 and pH 3.0 treatment groups, the soluble sugar contents decreased significantly in combined treatments of *P. oryzae* and SAR, compared with the non-inoculated treatments. The content of soluble sugar was significantly higher in SAR + *P.* treatment than in *P.* + SAR treatment at pH 5.0, pH 4.0, pH 3.0 and pH 2.0 groups (*P* < 0.05). 

In addition, stepwise regression showed that the disease index of rice leaves was more closely related to the content of lignin, and the regression equation of disease index and the metabolite content of rice leaves was ΔDI = −1.751 ΔLignin + 52.572 (*R*^2^ = 0.861, *P* < 0.001, *n* = 30).

### 2.5. Effects of SAR and P. oryzae on Rice Blast Disease Index 

The SAR treatments and timing of inoculation with *P. oryzae* significantly affected rice blast disease index (*P* < 0.001, Figure 5). Two-way ANOVA indicated that there was a strong interaction between SAR and inoculation time on rice blast disease index (*P* < 0.001). When the rice was inoculated after the simulated acid rain (SAR + *P.*) at pH 4.0, there was a significantly decreased disease index relative to the control (*P* < 0.05), while at pH 3.0 and pH 2.0 there was a significant increase in the disease index. The disease index was significantly increased under *P.* + SAR treatments at pH 4.0, pH 3.0 and pH 2.0, relative to the control, with the highest severity at pH 2.0 SAR treatment, followed by pH 3.0 and pH 4.0 (*P* < 0.05). In the combined treatments of *P. oryzae* and SAR, the disease index did not significantly vary at pH 5.0 (*P* > 0.05), but, at pH 4.0, pH 3.0 and pH 2.0 groups, the disease indexes were significantly higher in *P.*+ SAR treatments than those in SAR + *P.* treatments (*P* < 0.05), with a difference of 28%, 28%, and 8%, respectively.

### 2.6. Relationships between Disease Index and Total Number of Leaves, Enzyme Activities, and Metabolites

Relationships between disease index and total number of leaves, enzyme activities, and metabolites are analyzed (Figure 6). The results showed that the total number of leaves is significantly negatively correlated with disease index (Figure 6A). The activities of PAL, POD and CHT in rice leaves also negatively correlated to the disease index, indicating these enzymes, as indicators of plant resistance to environmental stress, play a very important role in responding and regulating disease incidence of rice leaves (Figure 6B–D). Moreover, the significantly negative correlations can be found between the contents of the four metabolites (flavonoids, total phenol, lignin and soluble sugar) and the disease index (Figure 6E–H), while the lignin content showed a more relevant relationship with the disease index (*R*^2^ = 0.861, *P* < 0.001, Figure 6G).

## 3. Discussion

In our study, although acid stress inhibited growth of *P. oryzae* in the culture experiments (Figure 1), the disease index of rice leaves increased under some high-intensity acid rain treatments significantly, this might be because acid rain treatment was performed every three days and the leaf surface returned to neutral, which may suggest that inoculated *P. oryzae* in this state will not be continuously affected by the acid rain. Previous studies showed that the acid stress inhibits the growth of pathogens rather than promote the growth in vitro, but the incidence of plants disease significantly increased after acid rain treatment in the pot experiment [26], which is consistent with our findings. Our results suggest that acid rain did not only directly affected the pathogens on the leaves, but also might result in an oxidative stress to plant leaves, and decreased antioxidant capacity and reduced the disease resistance of the leaves, thus increasing the incidence of disease. However, on the other hand, there is a complex antioxidant system to remove the excessive accumulation of reactive oxygen species (ROS) induced by environmental stress in plants [18]. Antioxidant enzyme activities can serve as an indicator of plant tolerance to abiotic stress [28,29]. We found that single acid rain treatments increased the PAL and CHT activities in leaves. Generally, acid rain stress can increase [11], or decrease POD activity depending on the stress intensity and duration [10]. The changes of POD activity in our study were in agreement with previous reports which suggested that increases in POD activity were observed in rice leaves under SAR of pH 5.0, 4.0, 3.0, and a decrease under SAR of pH 2.0 [22]. Both inoculation treatments at pH 7.0 increased PAL, POD and CHT activities, indicating that *P. oryzae* also induced the defense system of plant. The effects of *P. oryzae* on rice seedlings that we observed were similar to most reports [30,31]. Interestingly, in our study, the disease-resistance-related enzyme activities in rice leaves varied in response to the different inoculation times of *P. oryzae* (Figure 4). The changes of the total number of leaves, enzyme activities and metabolism compounds in rice leaves under the SAR + *P.* treatment at pH 4.0 and pH 2.0 suggested that regulating effects of *P. oryzae* on leaves were limited by the acidity of simulated acid rain, which is consistent with previous findings [32]. The changes in metabolites content and enzyme activities were inhibited in the *P.* + SAR treatment relative to the SAR + *P.* treatment. In addition, we found that there was an antagonistic effect of acid rain and *P. oryzae* on the protective enzyme system in rice leaves under SAR + *P.* treatment at pH 4.0. These results suggest that moderate-intensity acid rain positively influenced the front-line defense against pathogen-induced injury. The acid rain improved the PAL activity of the plant and stimulated the production of phenolic compounds, forming a protective barrier, reducing the availability of water and nutrients to pathogens, speeding up the pathogen cell death and reducing the incidence of disease [31]. That is an important reason for the lower disease index in SAR + *P.* treatment than that in *P.* + SAR treatment.

Phenolic compounds are the main secondary compounds synthesized by plants, and include flavonoids, total phenols and lignin [33,34], and are mainly produced by the styrene-acrylic metabolism pathway [35]. These compounds play an important role in plant growth and reproduction, providing protection against pathogens and predators [36]. The present study showed that the single simulated acid rain treatment at pH 4.0 increased the contents of the flavonoids, total phenol and soluble sugar in rice leaves (Table 1). These results were in agreement with previous reports on the effects of acid rain on the metabolites in rice leaves [37]. Totally, the contents of the flavonoids, total phenol and soluble sugar showed the similar trend of an increasing first and then decreasing under the combined stress of *P. oryzae* and acid rain when the pH value of SAR decreased. These results suggest that there would be a likely acidity threshold (pH ≥ 4.0) for rice defense system to acid rain stress, which would not affect the normal physiological activities of plants, but could improve the sensitivity of the plants’ resistance to stress. Below this threshold, the metabolic system would be destroyed and plant resistance would be reduced [11]. The changes in the contents of the flavonoids, total phenol and soluble sugar indicated that the single stress of inoculation with *P. oryzae* or acid rain increased the antioxidant metabolites, but the combined treatment stress of them under high acidity of simulated acid rain decreased and even inhibited the metabolites in rice seedlings. The lower metabolites suggest that higher-acidity acid rain would result in a lower resistance to disease. In addition, lignin content showed the highest contribution rate in physiological and biochemical parameters of rice leaves to the disease index, likely because it may strengthen the cell wall and has an important role in protecting plants from mechanical attacks from pathogenic bacteria [38,39]. In our study, the more severe disease index in *P*. + SAR treatment might be ascribed to the disruption of plant’s physical barrier induced by infection of *P. oryzae* and exacerbated by acid rain stress. The reduction in the lignin content of rice leaves under the combined treatments of high-acidity simulated acid rain (pH 3.0 and 2.0) and *P. oryzae* indicated that these combinations reduced the plant’s physical defense capabilities. Previous studies have also shown that, during the infection of pathogenic bacteria, the lignin content of the host crop is positively correlated with crop disease resistance [39,40], which is consistent with the results of this present study. Overall, the success or failure of resistance may depend on the temporal coordination of plant-induced defense responses [37,41].

The occurrence of plant disease is the result of plant-pathogen-environment interactions [34]. In this experiment, the simulated acid rain at pH 4.0 in the pre-infection stage of *P. oryzae* had a negative effect on the disease index of rice, but the combined treatments under high-intensity acid rain significantly increased the rice disease index (Figure 5). Based on the observed effects of acid rain on rice disease index, we think there are two interconnected plausible interpretations. First, the acid rain-induced “stress effect” maximized the plant defense against pathogenic in a short term. Second, the negative effect of excess H^+^ seemed to be slightly stronger than the “stress effect”. When the acidity of acid rain is within a readily tolerated threshold of the plant, acid rain can up-regulate the rice defense system and indirectly improve the its resistance to the fungal disease [42,43]. However, when the acidity of acid rain exceeded the tolerance range of rice, the rice would receive the dual stresses of pathogen and acid rain, and the result was a dysfunctional protective barrier, damaged antioxidant system and increased success rate of pathogen infection (Figure 4). With decreasing pH of acid rain, the total number of rice leaves first increased and then decreased (Figure 3). The reason may be that acid rain enriched in N acts as a fertilizer, which then accelerates biomass accumulation in leaves [11]. The total leaf number is a key indicator for calculating the disease index, and it is significantly negatively correlated with the disease index (Figure 6A). Moreover, high-acidity simulated acid rain decreased the disease tolerance of plant [16,44], resulting in the increased disease index of rice at pH 3.0 and pH 2.0 SAR treatment. Shriner et al. (1978) found that the inoculation of *Pseudomonas syringae pv. Phaseolicola* after the simulated acid rain (pH 3.2) treatment resulted in aggravation of the cowpea disease index, but simulated acid rain treatment after the symptomatic stage resulted in alleviation of the cowpea disease index [24], which diverges from our experiment. This discrepancy might be attributable to the crop variety, acid rain acidity, and pathogen type [5]. For example, different plants (rice, soybean and *Elaeocarpus glabripetalus* seedlings) have different tolerance to acid rain stress [10,11,45], so do the pathogens [5]. Hence, the interaction of different plant-pathogens to environmental stress would be varied.

## 4. Materials and Methods 

### 4.1. Preparation of Simulated Acid Rain and Cultivation P. oryzae

In southern China, the pH of rainfall averaged 4.5 between 2005 and 2009, with a low of 3.04 [46]. It is estimated that with further economic development in South China, the pH of precipitation would gradually decrease, and therefore an extremely low pH level (i.e., pH 2.0) was considered in this study. A simulated acid rain was prepared by using a stock solution at pH 1.0 which was concentrated solution of H_2_SO_4_ and HNO_3_ in a ratio of 4:1 (v/v) by chemical equivalents [45]. The acid rain solution was then adjusted to pH 5.0, 4.0, 3.0 and 2.0, respectively, with deionized water. A control was set using deionized water with the pH of 7.0. 

Strain Guy11 was provided by the laboratory of plant pathology in Guangdong province. The strain with medium 28 °C purification was cultured for 7 d, then was moved to the potato dextrose medium PDA (Potatoes-200 g, agar-16 g, glucose-20 g, distilled water-1 L) and cultured for 15 d at 28 °C in light conditions. In this experiment we used hydrochloric acid (HCl) and sodium hydroxide (NaOH) to adjust the pH (2.0, 3.0, 4.0, and 5.0), with a solution of pH 7.0 as a control [26] The spores were washed using the sterile water and adjusted to spore suspension liquid concentrations of 3 × 10^5^ (spore)/mL using a blood cell counting plate. 

### 4.2. Plant Cultivation and Experimental Design

The experiment was conducted in the Teaching and Research Farm (23°10′ N, 113°22′ E) of the College of Natural Resources and Environment, South China Agricultural University in Tianhe, Guangzhou, China. Seeds of a rice (*Oryza sativa*) named *Huang hua zhan* were provided by the Rice Research Institute of Guangdong Academy of Agricultural Sciences. The seeds were disinfected with carbendazim, and washed with deionized water three times, then germinated in an incubator for 6 d in a constant temperature and light incubator (28 °C). When the length of radicle of the seeds was about 3 cm, they were transferred to plastic pots (upper diameter of 51 cm, basal diameter of 38.5 cm, and height of 29 cm) with nine seedlings per pot and filled with 25 kg of topsoil from a local paddy farm near the Teaching and Research Farm. Soil total nitrogen content was 1.12 g/kg, total phosphorus content was 0.71 g/kg, alkali-hydrolyzed nitrogen, available phosphorus and organic matter contents were 123.50 mg/kg, 152.52 mg/kg and 13.50 g/kg respectively. When the rice grew to four leaves, we started the experiment. Chemical fertilizers and pesticides were not used during the whole experiment.

The experiment followed a factorial design of two factors of SAR and *P. oryzae* treatments. SAR constituted five levels that were the control, pH 5.0, pH 4.0, pH 3.0, and pH 2.0 treatments, while *P. oryzae* treatments included two stages of pre-infection stage and symptomatic stage. The growth cycle of *P. oryzae* is divided into four stages, the pre-infection stage, the incubation stage, the penetration stage and the symptom stage [47], but the incubation stage and penetration stage are difficult to distinguish. Therefore, we chose to study the pre-infection and symptomatic stages. After growing for 15 d, seedlings were subjected to SAR and divided into three groups with different inoculation treatments: (1) a non-inoculated group, cultivated under varying acid rain levels without *P. oryzae*; (2) a SAR + *P.* group where *P. oryzae* was inoculated at 72 h after SAR began (pre-infection stage); and (3) a *P.* + SAR group, where acid rain treatments began at 72 h after *P. oryzae* was inoculated (symptomatic stage).

To inoculate *P. oryzae* to rice leaves, the seedlings were sprayed with a spore suspension of *P. oryzae* from the top and sides to ensure that all the leaf surfaces were covered with 50 mL of inoculum, and then were immediately wrapped in black plastic bag for 24 h to maintain humidity and promote the conidia infection [48]. Simultaneously, we applied the same doses of 0.25% gelatin as a control (CK) treatment. Three replicates were established for each of the combinations of the SAR and inoculated treatments. Two liters of SAR solution (pH 5.0, pH 4.0, pH 3.0, and pH 2.0) were sprayed on the leaves of rice seedlings for each pot using a sprinkler at 9:00 a.m. every three days from August 2018 to October 2018 (total 20 times in 60 days), and set pH 7.0 with deionized water as control. The total amount of SAR was determined by combining the surface area of the pot with the annual precipitation and acid rain frequency from 2013 to 2017 in Guangdong Province, China [49].

### 4.3. Determination of P. oryzae Infectivity Related Indicators

The colony diameter was measured using the cross method every three days at 9 am, for 15 days, and the growth rate was calculated as follows.
Growth rate = (average colony diameter/incubation time) × 100%

Fifty milliliters of spore suspension was used for every nine seedlings at inoculation time. Rice seedlings were developed 24 h after inoculation darkness moisturizing (25~31 °C conditions), and then were turned to cultivate in artificial climate chamber, with the culture temperature of 25~31 °C [50]. Then the incidence was investigated after seven days.

Melanin content (y, mg/g) of *P. oryzae* extraction and purification methods were based on a previous study [51]. The sample of 0.20 g hyphae of *P. oryzae* was mixed with 20 mL 1 mol/L NaOH. The liquid was kept in a constant temperature water bath at 100 °C for 5 h, after that the liquid was filtered and added into a test tube, with water as blank. The colorimetric measure was conducted at 400 nm using a visible light spectrophotometer to calculate the melanin content.

### 4.4. Enzyme Activity Related to Disease Resistance in Rice Leaves Measurement

PAL activity was determined using ultraviolet spectrophotometry. A sample of fresh rice leaves (0.30 g) was ground after an ice bath and then centrifuged (10,000 r/min) for 10 min at 4 °C with a 5 mol/L mercaptoethanoboric acid buffer (containing 0.1 g/L PVP) of pH 8.8. After that, 20 μL of supernatant was collected, and added to 780 μL of 0.1 mol/L boric acid buffer and 200 μL of 0.02 mol/L phenylalanine. The reaction solution was kept in a constant temperature water bath at 30 °C for 0.5 h. The change of optical density (OD 290) value was determined with an ultraviolet spectrophotometer (UV-1810, Shanghai, China). One activity unit expresses the tissue protein per mg. In each 1 mL reaction system, the absorbance value of 290 nm changed to 0.10 unit per min [31].

POD activity was determined by the guaiacol method. A sample of fresh rice leaves (0.30 g) was mixed with 4 mL of 50 mm phosphate buffer (pH 7.8) and ground after an ice bath. The liquid was then centrifuged (8000 r/min) for 15 min at 4 °C and the supernatant was collected. For each sample, 50 μL of the supernatants were mixed completely with 1 mL of 0.3% H_2_O_2_, 0.95 mL of 0.2 % guaiacol solution and 1 mL of phosphate buffer (pH 7.0). 50 μL PBS at pH 7.8 instead of the enzyme was used as a control. Each sample was measured every 30 s for 2 min and the POD activity (U per mg of protein) was determined at 470 nm [52]. 

CHT activity was determined using ultraviolet spectrophotometry. Fresh rice leaves (0.10 g) were ground after an ice bath and 1 mL of acetic acid extracting solution was placed into the centrifuge at 12,000 r/min for 20 min at 4 °C and the supernatant was collected. For each sample, 0.4 mL of the supernatants were completely mixed with 0.2 mL of acetic acid extracting solution (0.05 mol/L, pH 5.0) and 0.4 mL of colloidal chitin solution (1%). The liquid was stored at 37 °C water for 1 h and centrifuged at 5000 r/ min for 10 min. Then, the N-acetyl glucosamine content in the supernatant was measured. Saturated borax solution (0.2 mL) was added into 0.4 mL of supernatant and stored in a boiling water bath for 7 min. After that, 0.2 mL of glacial acetic acid and 0.2 mL of 1% paradimethylaminobenzal-dehyde (DMAB) solution were added after cooling and stored in 37 °C water for 15 min. The optical density of the solution was measured at 585 nm with an ultraviolet spectrophotometry (UV-1810, Shanghai, china). One enzyme activity unit represents the amount of enzyme required for 1 g of tissue to decompose chitin to produce 1 mg N-acetyl glucosamine in 1 h [31]. 

### 4.5. Metabolite Related to Disease Resistance Measurement

Fresh leaves were used to analyze flavonoid and total phenol contents using ultraviolet spectrophotometry. For the determination of flavonoid content, 0.10 g of fresh rice leaves were collected with a leaf puncher and placed in a centrifuge tube, and extracted with 10 mL acidified methanol (methanol: water: hydrochloric acid = 79:20:1) for 30 min. Then, the reaction system was used to determine the absorbance value at 305 nm by ultraviolet spectrophotometer [53]. 

Total phenol content was assayed in a mixed reaction system containing 7 mL of methanol, 1 mL of water, and 2 mL of hydrochloric acid to extract the reaction for 24 h. Then, the reaction system was used to determine the absorbance value at 300 nm by ultraviolet spectrophotometer [54]. 

Lignin content was determined using ultraviolet spectrophotometry. Rice leaves (0.20 g) were mixed with 500 μL of 1% glacial acetic acid and 20 μL of perchloric acid. The liquid was then kept in a constant temperature water bath at 80 °C for 40 min and oscillated every 10 min and then 500 μL BaCl_2_ was added to each sample, 20 μL of the supernatants were mixed completely with 980 μL of K_2_Cr_2_O_7_-H_2_SO_4_ solution. Then, the reaction system was used to determine the absorbance value at 280 nm by ultraviolet spectrophotometer [55].

Soluble sugar content was determined by anthrone colorimetry. Fresh rice leaves (0.10 g) were ground after an ice bath and 7 mL water were added into a centrifuge tube. The water was boiled for 20 min in a water bath, and then cooled. The liquid was then centrifuged (3500 r min^−1^) for 10 min at 4 °C and the supernatant was collected. For each sample, 50 μL of the supernatants were mixed completely with 4 mL anthrone. The reaction solution was kept in a constant temperature water bath at 40 °C for 15 min, and the colorimetric value was measured at 625 nm using a visible light spectrophotometer to calculate the soluble sugar content [56].

### 4.6. Disease Index

Using the Standard Evaluation System for Rice (SES) of the International Rice Research Institute (IRRI 2002), the disease incidence was investigated after *P. oryzae* was inoculated for a week, and the disease index (DI) was calculated using following equation.
DI = ∑ (diseased level leaf number × representative value)/(total leaf number × heavy disease representative value) × 100%.

### 4.7. Statistical Aanalysis 

We used repeated measures analysis of variance (RM ANOVA) to test for significant differences of the periodically measured indices (such as the growth rate of *P. oryzae*) among treatments over time in the cultivation experiment of *P. oryzae*. Two-way analyses of variance (ANOVA) followed by Least-way analyse Difference (LSD) test were conducted to detect the main and interactive effects of SAR (pH) and inoculation time on total number of leaves, plant enzyme activities, antioxidant compounds contents and the disease index. Stepwise regression analysis was performed for all enzymes to determine the most powerful predictors for the disease index. Linear regression analysis was used to investigate the correlation between total number of leaves, enzyme activities, metabolites and the disease index. All these analyses were performed in SPSS 19.0 (IBM Corp., Armonk, NY, USA). Graphs were generated using Origin 9.1 (Origin Lab, Northampton, MA, USA). 

## 5. Conclusions

The results of our study demonstrate that, as the acidity of stimulated acid rain increases, the severity of rice blast disease generally increases significantly, but there is an exception, for example, less disease occurred with SAR + *P*. treatment at pH 4.0. Moreover, the acid rain treatments and timing of inoculation with *P. oryzae* can take different consequences and effects on rice blast disease index, which was observed significantly higher in *P.* + SAR treatments than that in SAR + *P.* treatments at the lower pH acid rain groups. One of the key mechanisms which acid rain and *P. oryzae* affected disease index of rice leaves is to interactively induce, change and regulate the activities of key antioxidant enzymes (PAL, POD and CHT) and the content of metabolites (such as flavonoids, total phenol, lignin and soluble sugar) in rice leaves. However, further research is needed to investigate the impacts and mechanisms of acid rain on rice blast or other rice diseases. 

## Figures and Tables

**Figure 1 plants-09-00881-f001:**
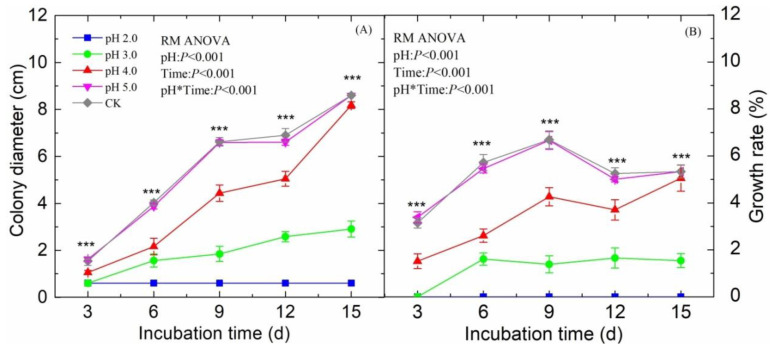
Colony diameter (**A**) and growth rate (**B**) of the *P. oryzae* in different cultivated periods under the acid treatments at pH 7.0 (CK), 5.0, 4.0, 3.0 and 2.0. In each panel, results of repeated-measures analysis of variance (RM ANOVA) are presented, with stars indicating the significance level (* *P* < 0.05; ** *P* < 0.01; *** *P* < 0.001).

**Figure 2 plants-09-00881-f002:**
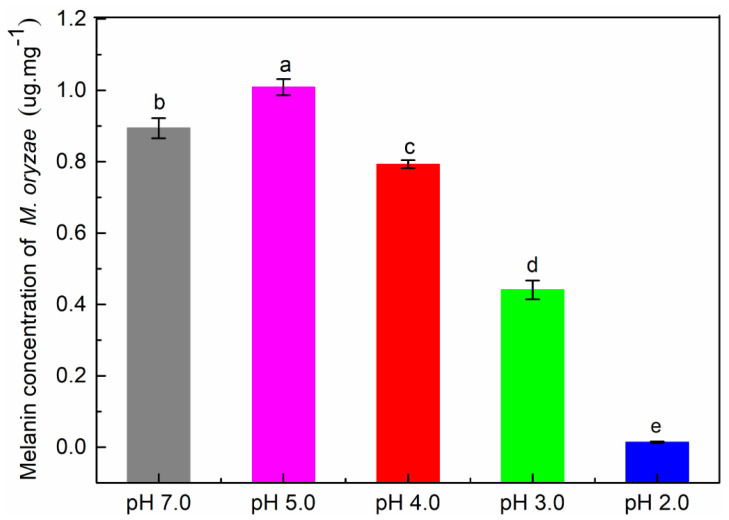
Melanin content of *P. oryzae* at different pH levels of in vitro culture treatments. Different letters above the bars indicate significant differences among treatments (*P* < 0.05). Values are the average ± standard error (*n* = 3).

**Figure 3 plants-09-00881-f003:**
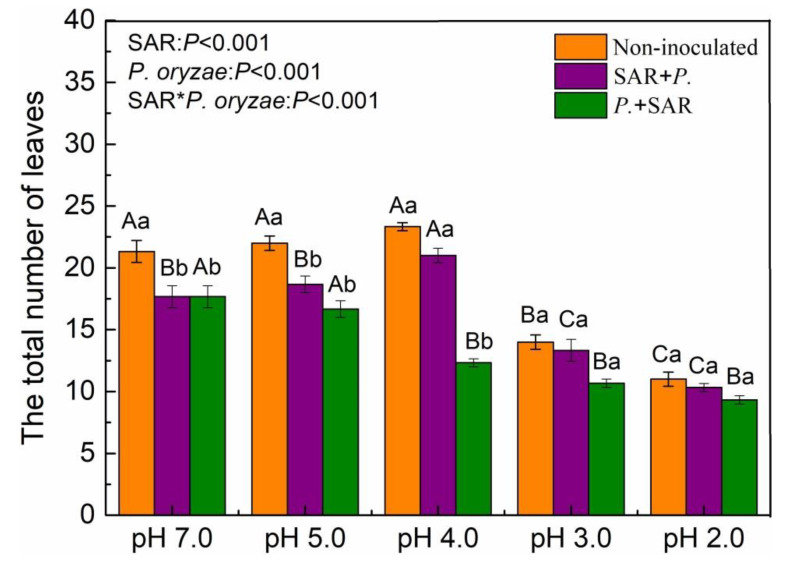
Total number of leaves under simulated acid rain (SAR) and inoculation of *P. oryzae* treatments. SAR + *P.*, spray acid rain at pre-infection stage of *P. oryzae*; *P.* + SAR, spray acid rain at symptomatic stage of *P. oryzae*. Lowercase letters indicate significant differences (*P* < 0.05) among treatments of the same SAR (pH) levels, while capital letters represent significant differences among different SAR (pH) levels for the same treatments (*P* < 0.05). Values are mean ± standard error (*n* = 9).

**Figure 4 plants-09-00881-f004:**
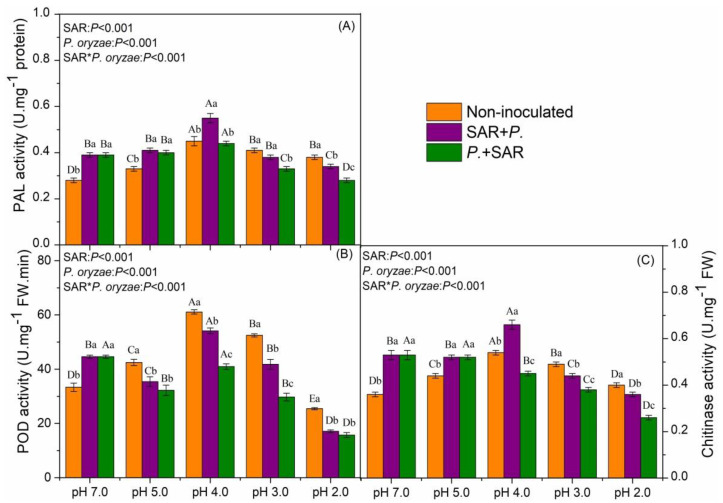
Phenylalanine ammonialyase (**A**; PAL), peroxidase (**B**; POD) and chitinase (**C**; CHT) enzyme activities of rice leaves under SAR and inoculated of *P. oryzae* treatments. SAR + *P.*, spray acid rain at preinfection stage of *P. oryzae*; *P.* + SAR, spray acid rain at symptomatic stage of *P. oryzae*. Lowercase letters indicate significant differences (*P* < 0.05) among treatments of the same SAR (pH) levels, while capital letters represent significant differences among different SAR (pH) levels for the same treatments (*P* < 0.05). Values are the average ± standard error (*n* = 3).

**Figure 5 plants-09-00881-f005:**
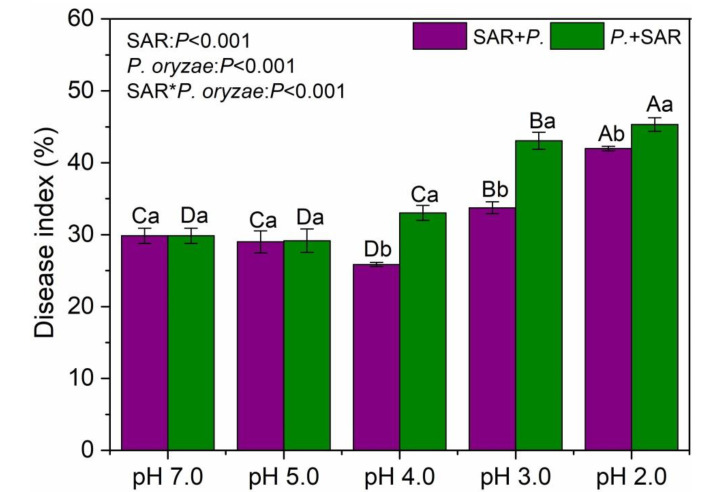
Disease index of rice leaves under different SAR levels. SAR + *P.*, spray acid rain at pre-infection stage of *P. oryzae*; *P.* + SAR, spray acid rain at symptomatic stage of *P. oryzae*. Lowercase letters indicate significant differences (*P* < 0.05) among treatments of the same SAR (pH) levels, while capital letters represent significant differences among different SAR (pH) levels for the same treatments (*P* < 0.05). Values are the average ± standard error (*n* = 3).

**Figure 6 plants-09-00881-f006:**
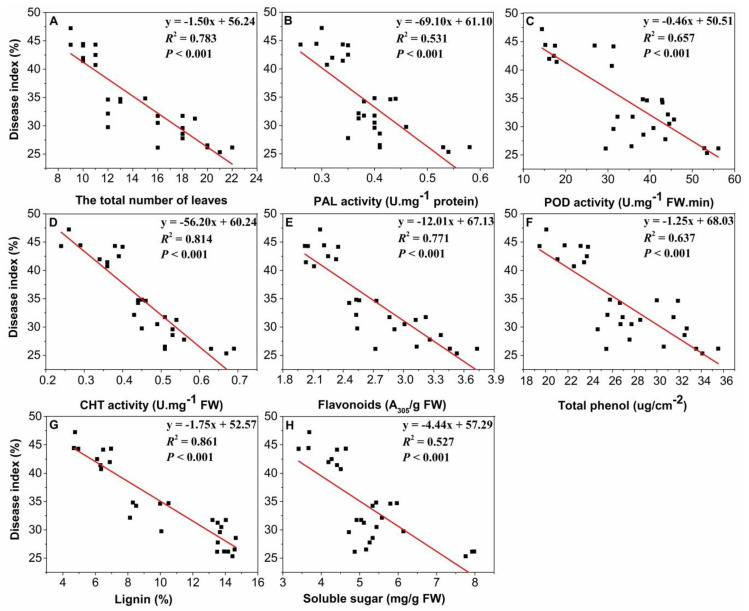
Relationships between disease index and (**A**) total number of leaves, (**B**–**D**) enzyme activities, and (**E**–**H**) metabolites. The regression equation, *R^2^*, and their significance level (*P*) are shown.

**Table 1 plants-09-00881-t001:** Flavonoids, total phenol, lignin and soluble sugar content in rice leaves under SAR and inoculation treatments. SAR + *P.*, spray acid rain at pre-infection stage of *P. oryzae*; *P.* + SAR, spray acid rain at symptomatic stage of *P. oryzae*. Capital letters represent significant differences among different SAR (pH) levels for the same treatments (*P* < 0.05), while lowercase letters indicate significant differences (*P* < 0.05) among treatments of the same SAR (pH) levels. Values are the average ± standard error (*n* = 3).

SAR (pH)	Inoculation (I)	Flavonoids (A_305_/g FW)	Total Phenol (μg/cm^−2^ FW)	Lignin (% DW)	Soluble Sugar (mg·g^−1^ FW)
CK	Non-inoculated	1.90 ± 0.05Cb	23.41 ± 0.36Db	18.41 ± 0.38Aa	3.00 ± 0.02Db
	SAR + *P.*	3.13 ± 0.07Ba	27.60 ± 0.52Ca	13.61 ± 0.07Ab	5.27 ± 0.10Ba
	*P.* + SAR	3.13 ± 0.07Aa	27.60 ± 0.52Ba	13.61 ± 0.07Ab	5.27 ± 0.10Ba
5.0	Non-inoculated	2.87 ± 0.04Ab	29.74 ± 0.45Bb	17.55 ± 0.45Aa	5.38 ± 0.02Ba
	SAR + *P.*	3.24 ± 0.07Ba	31.54 ± 0.54Ba	14.41 ± 0.19Ab	5.14 ± 0.12Ca
	*P.* + SAR	2.83 ± 0.06Bb	25.69 ± 0.66Cc	13.46 ± 0.13Ab	4.88 ± 0.09Bb
4.0	Non-inoculated	3.01 ± 0.06Ab	32.29 ± 0.44Ab	17.03 ± 0.20Aa	6.22 ± 0.50Ab
	SAR + *P.*	3.57 ± 0.08Aa	34.39 ± 0.59Aa	14.18 ± 0.15Ab	7.89 ± 0.07Aa
	*P.* + SAR	2.61 ± 0.06Cb	31.53 ± 0.80Ab	10.18 ± 0.16Bc	5.97 ± 0.10Ab
3.0	Non-inoculated	2.49 ± 0.08Ba	28.01 ± 0.27Ca	10.35 ± 0.36Ba	6.45 ± 0.07Aa
	SAR + *P.*	2.51 ± 0.02Ca	26.01 ± 0.34Ca	8.32 ± 0.11Bb	5.45 ± 0.07Bb
	*P.* + SAR	2.16 ± 0.10Cb	23.17 ± 0.36Ca	6.60 ± 0.19Cc	4.52 ± 0.07Cc
2.0	Non-inoculated	2.32 ± 0.05Ba	24.55 ± 0.52Da	8.10 ± 0.12Ca	4.65 ± 0.09Ca
	SAR + *P.*	2.20 ± 0.09Ca	22.75 ± 0.84Da	6.44 ± 0.23Cb	4.29 ± 0.06Cb
	*P.* + SAR	2.14 ± 0.05Ca	20.41 ± 0.67Db	4.19 ± 0.08Dc	3.60 ± 0.09Dc

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
