# Peer review of "Acid Rain Increases Impact of Rice Blast on Crop Health via Inhibition of Resistance Enzymes"

_plants, 2020, doi:10.3390/plants9070881_

Round 1
Reviewer 1 Report
I find these results preliminary. Improving the experimental design introducing other cultivars of Oryza sativa, to understand if the pathogenicity might be attributable to the crop variety, the manuscript could reach a very good level.
Also, the manuscript contains numerous errors and inaccuracies, which I report below.
For all these reasons I believe that the work is not finished and the results support, by the hair the conclusions, so I do not think it can be published in its current form
Line 23: stuyd --> study
Line 32: resulting in an increase in disease index--> increasing in disease index
Line 34: a key plant-pathogen system --> on the plant-pathogen interaction
Line 41: coloize --> colonizes
Line 60: polyphenlic compounds --> polyphenolic compounds
Line 64: of Pseu-domonas. phaseolicola--> caused by Pseudomonas phaseolicola
Line 203: Content--> content
Line 205: Mention first capital letters and then lowercase letters, as reported in Table 1 sequence.
Table 1: There are several errors in the table:
- In the last column, the data probably refers to the content of soluble sugars and not to phenols.
- It is necessary to report, such as the data relative to soluble sugar content was expressed (unit of measure).
- It is necessary to report, in each column, whether the value refers to the fresh weight or dry weight.
Line 252: environment --> environmental
Line 319: insert in reference Shriner et al. (1978)
Line 320: Pseu-domonas syringae pv. Phaseolicola--> Pseudomonas syringae pv. phaseolicola
Line 339: HCLàHCl
Line 349: 3cm--> 3 cm
Line 350: 51cm, basal diameter of 38.5cm--> 51 cm, basal diameter of 38.5 cm
Line 362: group--> groups
Line 387: was-->were
Line 404: of enzyme--> of the enzyme
Line 443-448: format the paragraph
Line 452: over time--> overtime
Line 470: Flavonoids-->flavonoids
Line 471: on the rice blast disease--> on rice blast disease
Reviewer 2 Report
This manuscript was described about the effect of acid rain on rice blast disease.
To summarize this manuscript from the viewpoint of plant pathology, acid rain treatments increased disease index with the exception in the pretreatment with mild acid rain (pH 4.0). This tendency was inconsistent with the result of the concentration-dependent growth inhibitory effect of blast fungus. It is considered that the factor that explains this contradiction is the inoculation condition. Acid rain treatment is performed every three days. After the treatment the leaf surface immediately returns to neutral. If the authors inoculated in that state, it will not affect the pathogenicity of blast fungus. Therefore, this paper only evaluated the effects of acid rain on rice physiology, not the effects on pathogens.
Discussions should be modified accordingly.
Specific comments
The name of pathogen should change from Magnaporthe oryzae to Pyricularia oryzae
(doi: 10.5598/imafungus.2016.07.01.09)
What is the definition of acid rain? (pH level or concentration of H2SO4 and HNO3)
What is CHT? (in Figure 4, described as chitin)
Round 2
Reviewer 1 Report
I suggest the manuscript for publication in present form.
Reviewer 2 Report
The revised manuscript is totally improved.